# Differential Synonymous Codon Selection in the B56 Gene Family of PP2A Regulatory Subunits

**DOI:** 10.3390/ijms25010392

**Published:** 2023-12-27

**Authors:** Gabriel Corzo, Claire E. Seeling-Branscomb, Joni M. Seeling

**Affiliations:** 1Department of Biology, Hofstra University, Hempstead, NY 11549, USA; gcorzo1@pride.hofstra.edu; 2Department of Biochemistry and Cell Biology, Stony Brook University, Stony Brook, NY 11794, USA; claire.seelingbranscomb@stonybrook.edu

**Keywords:** codon usage bias, synonymous, protein phosphatase 2A, B56 regulatory subunit

## Abstract

Protein phosphatase 2A (PP2A) functions as a tumor suppressor and consists of a scaffolding, catalytic, and regulatory subunit. The B56 gene family of regulatory subunits impart distinct functions onto PP2A. Codon usage bias (CUB) involves the selection of synonymous codons, which can affect gene expression by modulating processes such as transcription and translation. CUB can vary along the length of a gene, and differential use of synonymous codons can be important in the divergence of gene families. The N-termini of the gene product encoded by B56α possessed high CUB, high GC content at the third codon position (GC3), and high rare codon content. In addition, differential CUB was found in the sequence encoding two B56γ N-terminal splice forms. The sequence encoding the N-termini of B56γ/γ, relative to B56δ/γ, displayed CUB, utilized more frequent codons, and had higher GC3 content. B56α mRNA had stronger than predicted secondary structure at their 5′ end, and the B56δ/γ splice variants had long regions of weaker than predicted secondary structure at their 5′ end. The data suggest that B56α is expressed at relatively low levels as compared to the other B56 isoforms and that the B56δ/γ splice variant is expressed more highly than B56γ/γ.

## 1. Introduction

The B56 gene family encodes highly conserved regulatory subunits of the serine/threonine protein phosphatase 2A (PP2A). Yeast and human B56 sequences are approximately sixty percent identical. B56 proteins share a conserved central core domain but have divergent N- and C-termini which play a role in isoform specificity. B56 isoforms modulate canonical Wnt signaling; most B56 isoforms inhibit, whereas B56ε is required for, Wnt signaling [1,2,3]. Casein kinase Iδ and Iε, both activators of Wnt signaling, dissociate PP2A from the β-catenin degradation complex that inhibits Wnt signaling [4]. The vertebrate B56 gene family is likely the result of three duplications of a progenitor B56 gene and multiple gene losses, giving rise to five B56 loci [5]. The gene family consists of three B56-1 subfamily genes (B56α, B56β, and B56ε) and two B56-2 subfamily genes (B56γ and B56δ).

Because of the degeneracy of the genetic code, eighteen of twenty amino acids, all but methionine and tryptophan, are encoded by more than one codon. These synonymous codons, however, are not created equal, as they are not used at the same frequency. In fact, different species use different synonymous codons at different frequencies [6]. The differential use of synonymous codons is termed codon usage bias (CUB). The synonymous codon used can affect gene expression through several mechanisms, including transcription, translation, or post-translationally [7,8]. With regard to translation, this may be due to factors such as the abundance of corresponding charged tRNAs and mRNA secondary structure [8,9]. Interestingly, it has been shown that CUB can predict protein–protein interactions, as codon usage of genes encoding interacting proteins coevolve with one another [10]. Historically, studies of cancer-associated mutations have focused on mutations that change the encoded protein. However, recent studies have found that synonymous mutations play an important role as well, as they are predictive of cancer prognoses, presumably due to their effects on gene expression [11].

There are typically variations in CUB throughout a gene sequence, resulting in an intragenic codon landscape [9]. The presence of rare or frequent codons can modulate the rate of translation initiation and/or elongation, with translation rates slower with rare as compared to frequent codons. Rare codons in the first 30 to 50 codons of an mRNA, the ramp, may temporarily slow elongation, leading to extended regions between ribosomes. These regions lead to proper ribosome spacing and reduce the formation of ribosome traffic jams, thereby promoting higher rates of translation further downstream [12]. Rare codons in the ramp may also promote protein folding, for example, since the ramp is similar in size to the exit tunnel, which can hold approximately 40 amino acids and is sufficient in size to allow the formation of α-helices [8]. The strength of secondary structure at the 5′ end of the coding region affects translation, as strong mRNA secondary structures reduce the rate of translation [8]. An analysis of deviations from local free energy (LFE) of mRNA coding regions across a broad range of species defined the ramp as having four distinct regions [9]. It was found that in eukaryotes, the first approximately one to twenty codons have weaker than predicted secondary structure, followed by a transition at on average codon seventeen to moderately stronger than predicted secondary structure, which continues until immediately upstream of the stop codon, where mRNA again possesses weaker than predicted secondary structure [9]. In addition, a peak of stronger than predicted secondary structure is frequently present at the transition from weak to moderately strong secondary structure at the 5′ end of mRNA, at codons ten to twenty-three [9].

Gene duplication is an early step in the formation of a gene family [13]. After duplication, one copy of a gene typically maintains the original function, while the other copy is less evolutionarily constrained and may gain new functions due to mutations. Mutations that change the encoded amino acid are highly studied; however, CUB is another means by which members of a gene family may diverge from one another. CUB has been examined in several gene families, including ras, calmodulin, and polypyrimidine tract binding protein (PTBP) [14,15,16]. K-ras has a less significant role in cancer than H-ras, as K-ras possesses a large number of rare codons, and more mutations are required to increase K-ras expression to the level required to promote cancer [14]. Humans possess three calmodulin genes, each of which encodes identical proteins; however, CUB, as well as the presence of distinct regulatory DNA sequences, may lead to their differential tissue expression [15]. A high percentage of guanine or cytosine present at the third position in a codon, termed GC3, is an indicator of CUB and is associated with increased gene expression [16]. The increased gene expression may be due to factors such as methylation and RNA stability [17,18]. PTBP is present as a three-gene family in humans; high GC3 content in *PTBP1* correlates with high expression levels, while a low GC3 percent in *PTBP2* and *PTBP3* correlates with low expression levels [16]. Therefore, CUB plays a role in the divergence of gene family members.

In our previous study of the evolution of the B56 gene family through the analysis of nonsynonymous and synonymous substitutions, we found low synonymous substitutions in sequences encoding the termini of B56 proteins, suggesting that these termini experienced selection to maintain function [19]. As CUB can play an important role in gene expression, we further examined CUB in vertebrate B56 genes and the implications it may have on B56 gene expression. Several mechanisms were used to measure CUB including effective number of codons (Nc), Codon Adaptation Index (CAI), the percent of the third position of a codon containing guanine or cytosine (GC3%), and deviations of native mRNA secondary structure from structural predictions based on LFE, atypical RNA secondary structure, was found in B56α, and B56γ/γ and B56δ/γ splice variants showed differential CUB and secondary structure, suggesting that CUB plays a role in the regulation of B56 isoform gene expression.

## 2. Results and Discussion

### 2.1. Nc

Nc quantitates the number of different codons used in a protein-encoding nucleotide sequence and can vary from 20, which signifies high bias, to 61, which signifies low bias. An Nc analysis was carried out to further analyze CUB in vertebrate B56 genes. To characterize the intragenic codon landscape, Nc values were calculated for complete B56 gene sequences, as well as for each domain separately. Nc values of less than or equal to 35 are considered to possess CUB [20]. Nc values for the N-termini of B56α and B56γ satisfied this criterium and exhibited CUB, as their Nc values were 34 and 30, respectively (Figure 1A). Domains within each of the other isoforms had Nc values relatively similar to one another. We previously identified a highly expressed mixed-isoform splice variant of B56γ, B56δ/γ [21]. Evidence suggests that B56γ and B56δ arose from the duplication of a B56γ/B56δ progenitor gene [21]. The B56δ/γ splice variant has an N-terminal domain approximately 70% identical to the N-terminal domain of a frequently sequenced B56δ splice variant and a core region identical to B56γ. B56δ/γ and B56γ/γ are differentially expressed; for example, B56δ/γ is expressed more highly than B56γ/γ in early *Xenopus laevis* development [21]. The sequences encoding the N-termini of B56δ/γ were compared to both B56δ and B56γ/γ, as comparisons between B56δ/γ and B56δ may characterize mutations that occurred post-B56γ/B56δ gene duplication. The Nc values for B56δ/γ and B56δ N-termini encoding domains were 57 and 44, respectively, as compared to 30 for B56γ/γ; therefore, of these isoforms, only B56γ/γ possessed CUB (Figure 1B).

To further examine the intragenic codon landscape in the B56 gene family, the Nc value for each domain was normalized to the Nc value of its corresponding complete B56 gene sequence, as both intragenic and intergenic CUB exists [20]. It was found that the N-termini of B56α and B56γ had Nc values approximately 40% lower than their corresponding whole-gene sequences, whereas the differences for the other domains ranged from 2% higher to 17% lower than their corresponding whole-gene sequences, highlighting the CUB that was present in B56α and B56γ N-termini encoding domains (Figure 1C). In examining B56γ splice variants, it was found that the normalized Nc value for the domain encoding the N-termini of B56γ/γ was 45% lower than their corresponding whole-gene sequences, whereas the normalized Nc values for B56δ and B56δ/γ N-termini were zero, highlighting that CUB was only present in B56γ/γ N-termini encoding domains (Figure 1D). The data suggest that there has been selection for specific codons in the N-termini of both B56α and B56γ and that there was CUB in the N-termini of B56γ/γ but not B56δ/γ or B56δ. However, the low Nc value of B56γ/γ N-termini may not be highly informative due to their short lengths (32 codons on average), as others have excluded similarly short sequences in their analyses [22]. Nc values indicate whether codon bias is present but make no determination with regard to the direction of that bias. Comparisons of CAI values among the B56 isoforms were made to determine if the observed CUB favored rare or frequent codons.

### 2.2. CAI

CAI quantitates the use of rare/frequent codons in a protein-encoding nucleotide sequence. As each species may have a different codon usage pattern, the rarity of a codon is determined from a set of highly expressed genes in a given species and is summarized in a species-specific codon usage table [23]. The frequency of each codon per one thousand codons from this table is then used to calculate CAI values for gene sequences. A low CAI value results if a gene sequence contains a high number of rarely used codons, suggesting that a gene is expressed at low levels. Increasing CAI values, up to a value of one, correlate with genes possessing higher numbers of frequently used codons, suggesting that a gene is expressed at high levels. Due to gene differences in nucleotide content and the encoded amino acids, normalized CAI values are typically used in determining the significance of CAI values [23]. This normalization is performed using expected CAI (eCAI) values, which are average CAI values calculated from five hundred sets of synonymous sequences with similar GC content as the query sequence [24].

Percent differences between CAI and eCAI values were calculated and compared among domains within each B56 isoform. In these analyses, positive percent differences between CAI and eCAI signify the presence of frequently used codons, whereas negative values signify the presence of rare codons. The N-termini were the only domains in which large differences between CAI and eCAI values were found (Figure 2A,B). Specifically, the CAI value of B56α N-termini encoding sequences was 13% lower than its eCAI value, and B56β had a CAI value 8% higher than its eCAI value, while percent differences between CAI and eCAI values with all other B56 isoform N-termini were within 4% (Figure 2A). CAI was 3% higher than eCAI for B56γ/γ and B56δ N-termini, while it was 4% lower for B56δ/γ; therefore, B56γ/γ and B56δ N-termini possessed more frequently used codons than B56δ/γ (Figure 2B).

The 5′ end of a gene’s coding region plays an €mportant role in regulating gene expression [12,25]. Reduced translation rates may promote proper ribosomal spacing and high levels of expression due to rare codons in the ramp [8]. Studies in prokaryotes have shown that the extreme 5′ end, approximately ten codons, plays a crucial role in regulating the rate of translation. Rare codons are often present in the 5′ end, which suggests that rare codons directly reduce local translation efficiency. In the analysis of B56 isoform CAI values, the most salient feature found was that the first ten codons of B56α mRNA had particularly low CAI values, e.g., contained rare codons (G. Corzo and J.M. Seeling, Hofstra University, Hempstead, NY, CAIcal output graphs, 2023). Therefore, CAI analyses were performed using the first ten codons of each B56 isoform’s coding region. The first ten codons of B56α, and B56δ/γ as compared to B56γ/γ, displayed stronger differences between CAI and eCAI than with the entire N-termini. In particular, CAI was 47% lower than eCAI for B56α, and the percent difference of CAI versus eCAI for B56δ/γ was 12% lower, while it was 5% higher for B56δ, and CAI and eCAI were equal for B56γ/γ (Figure 2C,D). Thus, the first ten codons of B56α and B56δ/γ both possess a higher-than-expected abundance of rare codons. The abundance of rare codons in sequences encoding B56δ/γ N-termini and specifically the first ten codons suggests that there has been selection for rare codons in B56δ/γ and/or frequent codons in B56δ post-B56γ/B56δ gene duplication. The presence of rare codons in the ramps of B56α and B56δ/γ suggests that initial translation rates may be low to allow for proper ribosome spacing, potentially leading to higher translation rates further downstream, suggestive of higher expression levels than other B56 isoforms. This correlates with expression data from early *Xenopus laevis* development, which found that B56δ/γ was more highly expressed than B56γ/γ, although additional studies would be necessary to confirm this, and it is likely that other factors may be in play as well [7].

Some studies have found that the presence of rare codons in the 5′ coding region correlated with high expression rates, not due to the presence of rare codons in and of themselves, but due to a resultant weak secondary structure [25,26]. For example, through the analysis of genes present in hundreds of bacterial genomes, one group found that weak secondary structure, rather than the presence of rare codons directly, was the determining factor in translation efficiency [26]. Another group tested thousands of variants of codons 2–11 of a reporter mRNA in *E. coli*, maintaining amino acid identity, and found that the effect of rare codons was indirect in reducing local translational efficiency, with the direct effect being due to weak secondary structure [27]. To further examine potential regulation of B56 gene expression through CUB, GC3% of B56 isoform coding sequences were examined.

### 2.3. GC3%

In examining GC3 percent values, it was found that they were similar between the different domains of each isoform except for B56α, in which the domain encoding the N-termini possessed a GC3% value approximately two-fold greater than every other domain (Figure 3A). CAI had shown that this B56α domain displayed rare codons, which is not expected for a domain with high GC3 content; however, the GC3 codons present in this domain were extremely rare, resulting in low CAI values (G. Corzo and J.M. Seeling, Hofstra University, Hempstead, NY, CAIcal output graphs, 2023). For B56γ splice variants, the region encoding the N-termini of B56γ/γ and B56δ had GC3 percent values 70% and 93% higher, respectively, than that of B56δ/γ (Figure 3B). B56δ/γ’s low GC3% and negative CAI values both correlate with rare codons.

GC content varies within genomes due to the existence of isochores, which are long regions of DNA that have distinct GC content [28]. Due to the presence of GC isochores, we examined GC3 values normalized to GC3 percent values of each gene. The domains of all isoforms were found to have GC3% values within 15% of the value of their whole-gene sequence, except for B56α and B56γ N-termini, in which GC3 percent values were 96% and 25% higher than their whole genes, respectively (Figure 3C). In comparing B56γ splice variants, the domains encoding B56γ/γ N-termini were 26% higher, B56δ N-termini were the same, and B56δ/γ N-termini were 22% lower than their whole-gene sequences, resulting in the dissimilarity of the percent differences between domains encoding B56γ/γ and B56δ/γ N-termini being 48% (Figure 3D). Therefore, the whole-gene normalized GC3 content of the domains encoding B56α and B56γ/γ N-termini was higher than other B56 isoforms, with B56α possessing rare codons and B56γ/γ N-termini possessing frequent codons. High GC3 content may also be the result of selection for strong secondary structure, and as total GC content is more predictive of secondary structure than GC3 content, percent GC values of B56 isoforms were calculated.

### 2.4. GC

Genes with high GC content are associated with high gene expression in mammals; this may be due in part to increased transcription of GC-rich regions [7]. It was found, as with GC3, that GC content was similar between the different domains of each isoform except for B56α, in which the domain encoding the N-termini possessed a GC% value approximately two-fold greater than every other domain (Figure 4A). For B56γ splice variants, the region encoding the N-termini of B56γ/γ and B56δ had GC percent values similar to B56δ/γ, as they were only 30–40% higher than that of B56δ/γ (Figure 4B).

As done with GC3 analyses, the percent difference of GC content between the sequences encoding the N-termini of B56 isoforms and their respective whole-gene sequences was calculated to normalize for the presence of GC isochores. The value for B56α was three- to six-fold higher than the other B56 isoforms (Figure 4C). In examining B56γ splice variants, the domains encoding B56γ/γ and B56δ N-termini were 22% and 11% higher, respectively, and B56δ/γ N-termini were 5% lower than their whole-gene sequences, resulting in the dissimilarity of the percent differences between domains encoding B56γ/γ and B56δ/γ N-termini of approximately 27% (Figure 4D). This suggests that sequences encoding the N-termini of B56α and B56γ/γ have stronger secondary structure than the other B56 isoforms and may be expressed at lower levels. In addition, data suggest that there has been selection for low GC3 and GC content in B56δ/γ and/or high GC3 and GC content in B56δ post-B56γ/B56δ gene duplication. To further examine the potential effects of CUB on B56 gene expression, we calculated the difference in LFE between B56 isoforms’ native mRNA secondary structure and predicted structures.

### 2.5. Local Folding Energy

CUB may affect gene expression through its influence on mRNA secondary structure; however, there is data that suggest that mRNA secondary structure may be selected for directly, rather than being solely a downstream effect of selection for CUB through GC content [9]. mRNA folding occurs in a localized fashion as transcription occurs; the use of sliding windows in local free energy calculations mimics this process and is therefore efficient at predicting RNA secondary structures present during translation [9]. The difference between the LFE of native B56 mRNA and that predicted by random RNA sequences encoding the same protein with the same nucleotide content and codon frequencies (∆LFEs) was calculated [9]. The typical pattern in eukaryotic organisms consists of weaker secondary structure than expected at the beginning and end of the coding region, with the initial weak secondary structure followed immediately by a depression of relatively stronger secondary structure than predicted, which is then followed by more moderate secondary structure through the remainder of the coding region [9]. In focusing on the 5′ end of B56 coding sequences, it was found that most B56 mRNAs (B56β, B56γ, and B56ε) follow the pattern exhibited by typical eukaryotic mRNAs, containing a short (one to three codons) domain of weak secondary structure followed by a region of relatively strong secondary structure (Figure 5B,C,E). B56α and B56δ mRNA did not follow the typical pattern, however, as they displayed stronger than predicted secondary structure in the first 51 or 2 codons of the coding region, respectively (Figure 5A,D). This region of B56α encompasses the region that possessed CUB (Figure 1A) and rare codons (Figure 2C), suggesting that the CUB resulted in the presence of rare codons, high GC3 content, and strong secondary structure. The lack of domains with weak secondary structure at their 5′ end suggests that B56α and B56δ will have a low translational initiation rate and therefore will be expressed at lower rates than other B56 isoforms.

With regard to the alternative splice forms of B56γ, it was found that B56δ/γ varied somewhat from the typical B56 isoform pattern as its ramp of weak secondary structure was up to eight-fold longer than the ramp of other B56 genes, as it extended through 25 codons, while it extended through three codons in B56γ/γ (Figure 5F,H). B56δ began with two codons with strong secondary structure, suggesting a slow rate of translational initiation (Figure 5G). The difference between the sequences encoding the N-termini of B56δ and B56δ/γ suggests that at the start of the coding region, there was selection for stronger than predicted secondary structure and low expression for B56δ while weaker than expected secondary structure and high expression for B56δ/γ. The extended weakly structured region in B56δ/γ may result in it being more accessible to the ribosome, and therefore, it may have higher rates of translational initiation and higher expression levels than the alternative B56γ splice variant B56γ/γ (Figure 5F–H).

### 2.6. Synonymous Mutations of B56 Isoforms in Cancer

As synonymous mutations play a significant role in cancer, most likely due to the modulation of gene expression, we examined whether B56 isoform mutations have been identified in human cancer. The Catalogue of Somatic Mutations in Cancer (COSMIC) was searched, and numerous cancer-associated synonymous mutations were found in the B56 isoform genes, comprising 17% to 27% of all cancer-associated mutations of each gene; the number of synonymous mutations was 28% to 55% of the number of missense mutations (Table 1). These values were comparable to the values determined through an examination of all genes present on COSMIC, in which it was reported that synonymous mutations made up 23% of all mutations and were 37% of the number of missense mutations [29]. The presence of a relatively large number of synonymous substitutions in B56 genes found in cancer suggests that these mutations modulate B56 gene expression through, for example, changes in mRNA secondary structure.

## 3. Materials and Methods

### 3.1. Identification of B56 Isoform Homologs

Each human B56 isoform protein was used as a query to search the National Center for Biotechnology Information (NCBI) vertebrate non-redundant (nr) protein sequence database (https://www.ncbi.nlm.nih.gov/protein/, last accessed on 17 June 2023) using Basic Local Alignment Search Tool protein (BLASTp, National Library of Medicine, Bethesda, MD, USA) version 2.12.0+ (https://blast.ncbi.nlm.nih.gov/Blast.cgi, last accessed on 17 June 2023) with the default parameters. The GenBank accession numbers used in the query were B56α: NP_006234.1, B56β: NP_006235.1, B56γ: NP_002710.2, B56δ: NP_006236.1, B56ε: NP_001269108.1, and B56δ/γ: XP_005267877.1. The sequences with the highest percent identity and percent query coverage in each species were retrieved. Sequences with gaps of sixty or more amino acids were excluded from the analyses. Vertebrate species that possessed each isoform were selected for further analyses. Sequences from each B56 isoform were aligned using Clustal Omega (European Bioinformatics Institute of European Molecular Biology Laboratory, Hinxton, United Kingdom, version 1.2.2) to verify their identity and the completeness of their sequence (https://www.ebi.ac.uk/Tools/msa/clustalo, last accessed on 2 June 2023). A given species was used only if it had a codon usage table on the Codon Usage Database (http://www.kazusa.or.jp/codon/, last accessed on 26 June 2022). The nucleotide sequence for each protein sequence was retrieved from the nucleotide database on GenBank (https://www.ncbi.nlm.nih.gov/nuccore, National Library of Medicine, Bethesda, MD, USA, last accessed on 17 June 2023). The GenBank accession numbers retrieved were B56α: NM_006243.4, B56β: NM_006244.4, B56γ: NM_002719.4, B56δ: NM_006245.4, B56ε: NM_001282179.3, and B56δ/γ: XM_005267820.1. Twenty-eight species contained all five isoform sequences of sufficient quality and were used in the CUB analyses. Of these species, eighteen had sequences of the B56γ splice variant B56δ/γ of sufficient quality. The eighteen species included *Homo sapiens*, *Pelodiscus sinensis*, *Cricetulus griseus*, *Mus musculus*, *Oryctolagus cuniculus*, *Rattus norvegicus*, *Bos taurus*, *Felis catus*, *Ovis aries*, *Aotus nancymaae*, *Chlorocebus sabaeus*, *Macaca fascicularis*, *Macaca mulatta*, *Microcebus murinus*, *Otolemur garnettii*, *Pan paniscus*, *Pan troglodytes*, and *Papio anubis.* The twenty-eight species included additionally *Gekko japonicus*, *Cavia porcellus*, *Microtus ochrogaster*, *Octodon degus*, *Camelus dromedarius*, *Equus asinus*, *Mustela putorius furo*, *Sus scrofa*, *Saimiri boliviensis*, and *Tupaia chinensis*.

### 3.2. CUB Calculations

Nc, eCAI, CAI, percent GC3, and percent GC were calculated using CAI calculator (https://ppuigbo.me/programs/CAIcal/, Universitat Rovira I Virgili, Catalonia, Spain, last accessed on 26 October 2023). CAI calculator uses the Ncw version of Nc [30]. eCAI values were calculated with population and confidence intervals of 90%. Percent differences between eCAI and CAI were used to determine the significance of CAI values.

### 3.3. RNA Local Free Energy Calculations

Changes in local folding energy (ΔLFE) between native and the average of twenty permutations of synonymous codons for a given B56 isoform were calculated using the DeltaLFE calculation demonstration code published on GitHub [9]. The code was modified to process the entire length of the input sequences and to provide x-intercepts in the output. A 40-nucleotide sliding window analysis with a step interval of 10 nucleotides using the standard genetic code was performed. Twenty-eight species were used in the analyses of the five B56 genes; eighteen species were used in the analyses of B56γ splice variants.

### 3.4. COSMIC Searches

COSMIC version 99, released 28 November 2023 (https://www.sanger.ac.uk/tool/cosmic/, Wellcome Sanger Institute, Hinxton, UK, last accessed 15 December 2023) was searched using the gene names of human B56 isoforms: B56α, *PPP2R5A*; B56β, *PPP2R5B*; B56γ, *PPP2R5C*; B56δ, *PPP2R5D*; and B56ε, *PPP2R5E*.

## 4. Conclusions

The N-termini of the gene product encoded by B56α possessed high codon bias, high GC3 content, high rare codon content, and stronger than predicted mRNA secondary structure. Of two B56γ splice variants, the sequences encoding the N-termini of B56γ/γ displayed CUB, utilized more frequent codons, and had higher GC3 content, while B56δ/γ displayed an extended region of weaker than predicted RNA secondary structure at their 5′ end. The data suggest that within the B56 gene family, B56α is expressed at relatively low levels and that B56δ/γ is expressed more highly than B56γ/γ. The abundance of synonymous mutations of B56 isoforms found in human cancer suggests that these mutations may play a role in cancer through the modulation of B56 isoform expression.

## Figures and Tables

**Figure 1 ijms-25-00392-f001:**
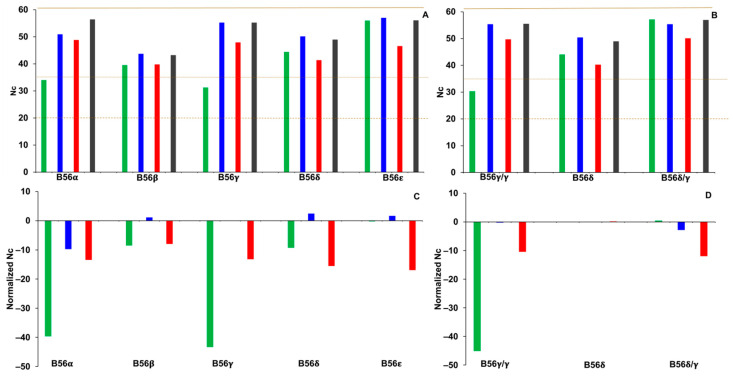
B56α and B56γ exhibited CUB (codon usage bias). Effective number of codons (Nc) values were calculated for B56 isoform domains. Green bars represent N-termini, blue bars represent the core region, red bars represent C-termini, and gray bars represent the entire coding region. (**A**,**B**) The dashed brown line marks the minimum Nc value, the dotted brown line demarcates the cutoff for CUB, and the solid brown line demarcates the maximum Nc value. (**C**,**D**) Nc values were normalized for B56 isoform domains by calculating the percent difference between each domain and its entire coding sequence. Twenty-eight species were used in the analyses of the five B56 genes; eighteen species were used in the analyses of B56γ splice variants.

**Figure 2 ijms-25-00392-f002:**
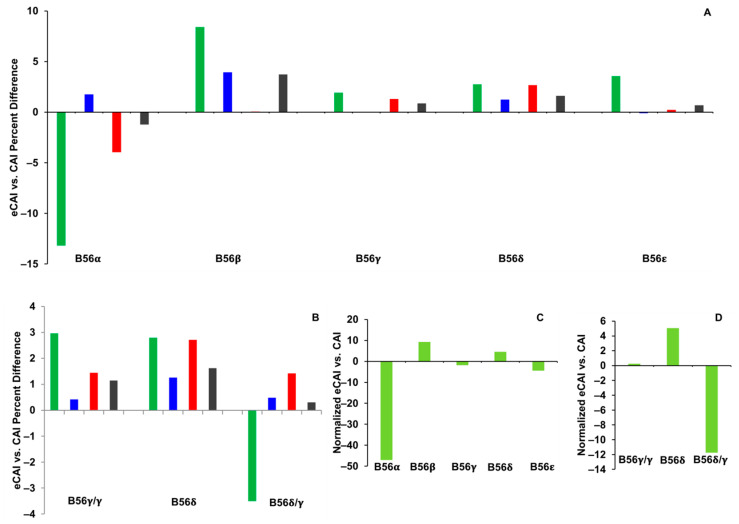
Rare codons were abundant in B56α and B56δ/γ. Codon Adaptation Index (CAI) and expected Codon Adaptation Index (eCAI) values were calculated for B56 isoforms. Percent differences between CAI and eCAI were plotted. Green bars represent N-termini, blue bars represent the core region, red bars represent C-termini, and gray bars represent the entire coding region. (**A**,**B**) CAI and eCAI values were calculated for B56 isoform domains. (**C**,**D**) CAI and eCAI values were calculated for the first ten codons of B56 isoforms. Twenty-eight species were used in the analyses of the five B56 genes; eighteen species were used in the analyses of B56γ splice variants.

**Figure 3 ijms-25-00392-f003:**
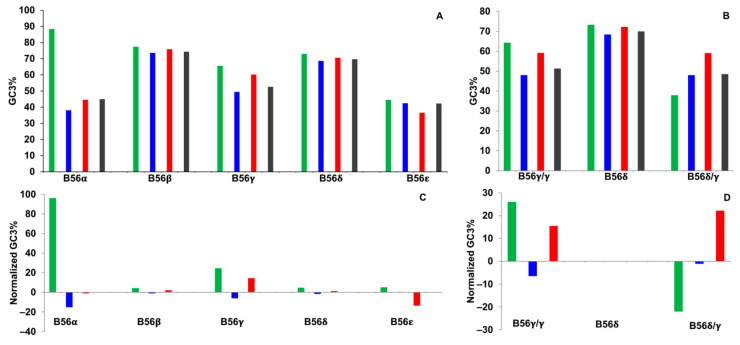
B56α possessed a high percentage of guanines and cytosines at the third position of codons (GC3%), and there was differential GC3% between B56γ alternative splice forms. GC3% values were calculated for B56 isoform domains. Green bars represent N-termini, blue bars represent the core region, red bars represent C-termini, and gray bars represent the entire coding region. (**A**,**B**) Percent GC3 values were plotted. (**C**,**D**) GC3% values were normalized for each B56 isoform by calculating the percent difference of each domain as compared to the entire coding sequence. Twenty-eight species were used in the analyses of the five B56 genes; eighteen species were used in the analyses of B56γ splice variants.

**Figure 4 ijms-25-00392-f004:**
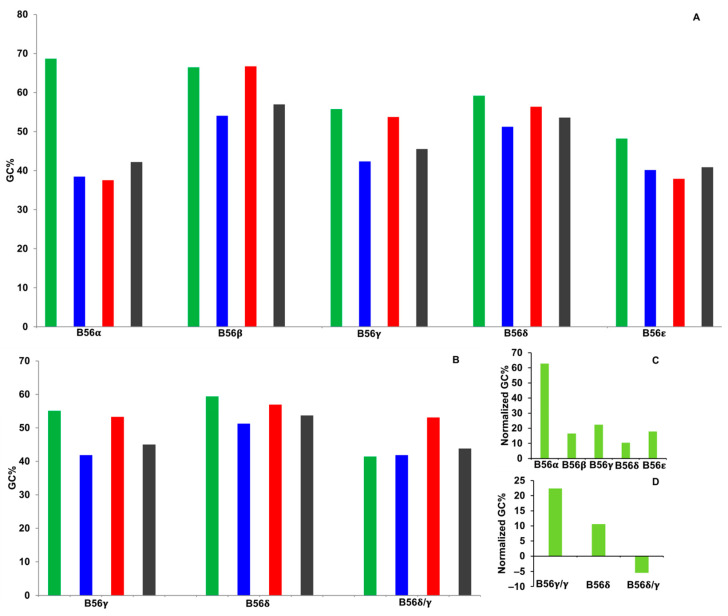
Guanine cytosine content (GC%) varied within and between B56 isoforms. GC% values were calculated for B56 isoform domains. Green bars represent N-termini, blue bars represent the core region, red bars represent C-termini, and gray bars represent the entire coding region. (**A**,**B**) Percent GC values were plotted. (**C**,**D**) GC% values were normalized for each B56 isoform by calculating the percent difference of each domain as compared to its entire coding sequence. Twenty-eight species were used in the analyses of the five B56 genes; eighteen species were used in the analyses of B56γ splice variants.

**Figure 5 ijms-25-00392-f005:**
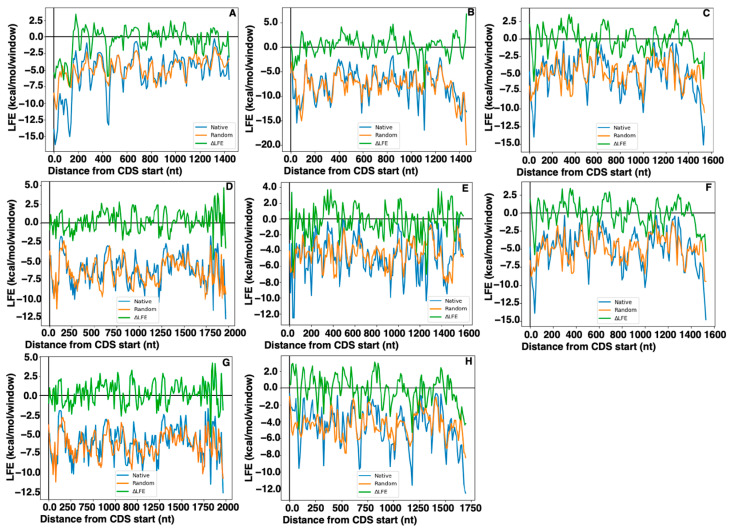
The 5′ ends of B56α and B56δ/γ coding regions displayed atypical RNA secondary structure. Differences between the local free energy (LFE) of native B56 mRNA and that predicted by random RNA sequences encoding the same protein with the same nucleotide content and codon frequencies (∆LFEs) were calculated for each of the B56 coding regions. LFE was plotted against the distance from the coding sequence (CDS) start in nucleotides (nt). (**A**) B56α, (**B**) B56β, (**C**) B56γ, (**D**) B56δ, (**E**) B56ε, (**F**) B56γ/γ, (**G**) B56δ, and (**H**) B56δ/γ. Twenty-eight species were used in the analyses of the five B56 genes; eighteen species were used in the analyses of B56γ splice variants.

**Table 1 ijms-25-00392-t001:** Abundance of synonymous B56 isoform mutations in cancer. Mutations in B56 isoforms were tabulated from the Catalogue of Somatic. Mutations in Cancer (COSMIC).

B56 Isoform	Abundance of Synonymous Mutations with Respect to All Mutations (%)	Abundance of Synonymous Mutations with Respect to Missense Mutations (%)
B56α	17	36
B56β	25	40
B56γ	16	28
B56δ	27	55
B56ε	18	30

## Data Availability

Data is contained within the article.

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
