# Peer review of "Differential Synonymous Codon Selection in the B56 Gene Family of PP2A Regulatory Subunits"

_ijms, 2023, doi:10.3390/ijms25010392_

Round 1

Reviewer 1 Report

Comments and Suggestions for Authors

Dear Editor,

Thank you for the opportunity to review this paper.

The paper presents very interesting data on the distinct patterns of synonymous codon usage in the B56 gene family of PP2A regulatory subunits. However, the manuscript lacks experimental verification.

1-   The resolution and font size of all the figures should be enhanced to improve their readability and quality. 

2-   What is the rationale for not examining the possible effects of codon usage bias on post-translational modifications or protein-protein interactions?

3-   The paper offers some clues into the differential codon usage bias, but does not explain the specific molecular mechanisms underlying this regulation.

4-  The paper does not explore the potential link between codon usage bias and disease states or pathophysiological conditions. Is there any evidence or hypothesis to support such a connection?

Reviewer 2 Report

Comments and Suggestions for Authors

The paper gives a detailed analysis of the codon usage of a gene family and provide meaningful conclusions from the results. Before publication, however, a few minor issues need to be addressed as described below.

Give formal definitions of CUB, Nc, CAI GC3.

Be specific about which codons are considered rare. Is there a threshold? Is the rareness specific to an organism or to  family of organisms?

L269-272: Beside giving s reference, a brief description of the method calculating the LFE would be helpful.

L 329: GitHub is the name of the software depository where the program used is, it is not the name of the program.

Round 2

Reviewer 1 Report

Comments and Suggestions for Authors

Dear editors, 

Thank you again for inviting me to review the paper entitled "Differential synonymous codon selection in the B56 gene family of PP2A regulatory subunits". All the suggested changes have been made satisfactorily. Therefore, I would like to recommend accepting this manuscript for publication in the International Journal of Molecular Sciences.

Sincerely,